# Treatment Strategies for Non-Small Cell Lung Cancer with Common *EGFR* Mutations: A Review of the History of EGFR TKIs Approval and Emerging Data

**DOI:** 10.3390/cancers15030629

**Published:** 2023-01-19

**Authors:** Julian A. Marin-Acevedo, Bruna Pellini, ErinMarie O. Kimbrough, J. Kevin Hicks, Alberto Chiappori

**Affiliations:** 1Division of Medical Oncology, Indiana University Melvin and Bren Simon Comprehensive Cancer Center, Indianapolis, IN 46202, USA; 2Department of Thoracic Oncology, Moffitt Cancer Center and Research Institute, Tampa, FL 33612, USA; 3Department of Oncologic Sciences, Morsani College of Medicine, University of South Florida, Tampa, FL 33620, USA; 4Division of Hematology and Oncology, Mayo Clinic, Jacksonville, FL 32224, USA; 5Department of Individualized Cancer Management, Moffitt Cancer Center, Tampa, FL 33612, USA

**Keywords:** *EGFR* mutations, EGFR TKIs, non-small cell lung cancer, targeted therapies, therapy sequencing

## Abstract

**Simple Summary:**

The management of non-small cell lung cancer with a common *EGFR* mutation has evolved over the past decades. While frontline use of second- or third-generation EGFR tyrosine kinase inhibitors (TKIs) is preferred over first-generation EGFR-TKIs, choosing the ideal agent depends on multiple factors (drug availability, physician comfort, specific *EGFR* mutation, presence of brain metastasis, etc.). Furthermore, defining subsequent therapies at the time of progression will rely on numerous variables (extent of disease, frontline EGFR TKI generation used, mechanism of resistance, etc.). Consequently, defining an optimal sequencing strategy is both, crucial and challenging. In this review, we present a detailed summary of evidence supporting the use of EGFR TKIs with or without other therapeutic approaches, outline available options at the time of disease progression, summarize investigational strategies, and suggest an approach to therapeutic sequencing in patients with common *EGFR* mutations.

**Abstract:**

The development of targeted therapies over the past two decades has led to a dramatic change in the management of *EGFR*-mutant non-small cell lung cancer (NSCLC). While there are currently five approved EGFR tyrosine kinase inhibitors (TKIs) for treating *EGFR*-mutant NSCLC in the first-line setting, therapy selection after progression on EGFR TKIs remains complex. Multiple groups are investigating novel therapies and drug combinations to determine the optimal therapy and treatment sequence for these patients. In this review, we summarize the landmark trials and history of the approval of EGFR TKIs, their efficacy and tolerability, and the role of these therapies in patients with central nervous system metastasis. We also briefly discuss the mechanisms of resistance to EGFR TKIs, ongoing attempts to overcome resistance and improve outcomes, and finalize by offering treatment sequencing recommendations.

## 1. Introduction

Sensitizing mutations in the epidermal growth factor receptor (*EGFR*) gene are one of the most common targetable genomic alterations in non-small cell lung cancer (NSCLC). These can be found in ~15% of lung adenocarcinomas in the United States and 22–64% of lung adenocarcinoma in Asian patients [1,2]. In the past two decades, five EGFR tyrosine kinase inhibitors (TKIs) have become commercially available for the management of advanced NSCLC with common *EGFR*-sensitizing mutations [i.e., *EGFR* exon 19 deletions or exon 21 mutations (L858R)]: erlotinib, gefitinib, afatinib, dacomitinib, and osimertinib (Table 1; Figure 1) [3,4,5,6,7,8,9]. These drugs exhibit distinct activity and safety profiles and are divided into first- (e.g., erlotinib, gefitinib), second- (e.g., afatinib, dacomitinib), and third-generation (e.g., osimertinib) TKIs. The second- and third-generation TKIs, afatinib and osimertinib, have demonstrated extended activity against some uncommon *EGFR* mutations [e.g., T790M (osimertinib), G719X, L861Q, or S768I (afatinib and osimertinib)] [10,11,12,13,14].

The efficacy of the EGFR TKIs has improved with each new generation. First- and second-generation EGFR TKIs have superior response rates (RRs) and progression-free survival (PFS) compared to platinum-doublet chemotherapy (i.e., cisplatin or carboplatin combined with either gemcitabine, pemetrexed, paclitaxel, or docetaxel) [3,4,5,6,15]. Dacomitinib and osimertinib confer improved PFS and overall survival (OS) rates compared to the first-generation EGFR TKIs [7,8,16,17]. Afatinib confers superior PFS and OS compared to platinum-doublet chemotherapy but did not improve OS compared to first-generation EGFR TKIs [6,18,19]. While first-generation EGFR TKIs have been compared head-to-head with second- and third-generation TKIs and have proven to be less effective, second- and third-generation EGFR TKIs have not been compared head-to-head in a prospective clinical trial. Therefore, it is unclear if these drugs confer the same outcomes for patients.

Despite EGFR TKIs improved efficacy over chemotherapy, drug resistance eventually occurs [20]. A wide array of genomic pathways activation and transcriptional remodeling have been reported as mechanisms of resistance [20]. These findings have prompted investigation of novel combinations and therapeutics. In this review, we discuss the data to support the use of EGFR TKIs alone or in combination, describe developing therapies for resistant disease, and propose a treatment sequencing strategy based on the available data.

## 2. Exploring the Use of EGFR TKIs

### 2.1. EGFR TKIs versus Platinum-Doublet Chemotherapy

In July 2002, gefitinib became the first EGFR TKI to be approved in the world, specifically in Japan, for advanced NSCLC [21]. Two years later, erlotinib was approved in the US for unselected patients with advanced NSCLC [22,23]. Nine years later it was approved in the first-line setting for the treatment of advanced NSCLC with an *EGFR* exon 19 del (ex19del) or L858R mutation based on findings of the EURTAC study [3,24]. This phase 3 trial demonstrated improvement in RR (64% vs. 18%) and PFS (9.7 months vs. 5.2 months; *p* < 0.0001) in the intention-to-treat patient population when compared to a platinum-doublet [3]. Erlotinib was better tolerated, and its use was associated with less serious adverse events (AEs) (6% vs. 20%) (Table 1) [3].

In 2009, the European Medicines Agency (EMA) approved gefitinib for the treatment of advanced *EGFR*-mutant NSCLC based on the phase 3 clinical trial IPASS [4,25]. Gefitinib showed improved RR (43% vs. 32%) and OS (18.8 vs. 17.4 months; *p* = 0.109) compared to chemotherapy in a population of light/never-smokers East Asians patients with advanced pulmonary adenocarcinoma [4,26]. Subgroup and post-hoc analyses of this study confirmed that gefitinib conferred a superior RR for all patients with *EGFR*-mutant NSCLC (71% vs. 47%). Specifically, the RR was 85% vs. 43% for patients with ex19del and 61% vs. 53% for the ones with L858R mutations [4,26]. Following IPASS, several landmark trials have confirmed the superiority of gefitinib over platinum-doublet in the first-line setting for patients with *EGFR*-mutant NSCLC in East Asian (NEJ-002, WJTOG3405) and White patients (IFUM) (Table 1) [27,28,29,30,31]. In 2015, the U.S. Food and Drugs administration (FDA) approved gefitinib for first-line treatment of metastatic NSCLC [32].

Afatinib was the first second-generation EGFR TKI approved by the FDA in 2013 [33]. It demonstrated improved PFS and RR compared to chemotherapy in the LUX-Lung 3 (PFS 11 vs. 7 months, *p* = 0.0004; RR 56% vs. 23%) and the LUX-Lung 6 (PFS 14 vs. 6 months, *p* < 0.0001; RR 67% vs. 23%) clinical trials [5,6,9,34]. Interestingly, in both trials the OS benefit from afatinib was seen among patients with *EGFR* ex19del mutations but not in those with L858R mutations (Table 1) [6]. Up to 49% of patients receiving afatinib developed serious toxicities, however, cessation of treatment due to side effects was more common among those receiving chemotherapy [5,6,9]. According to the retrospective RealGiDo study, afatinib can be dose-reduced to improve tolerability without affecting efficacy [35].

Osimertinib was initially approved for patients with *EGFR* T790M mutant-NSCLC who had progressed on or after EGFR TKI therapy [36]. The phase 3 AURA3 trial demonstrated improved RR (71% vs. 31%) and median PFS (10.1 vs. 4.4 months; *p* < 0.001) in those receiving osimertinib vs. platinum-based chemotherapy [37]. In this study, osimertinib also demonstrated activity against asymptomatic central nervous system (CNS) metastasis while showcasing a safe toxicity profile. Only 23% of patients experienced serious AEs compared to 47% of patients in the chemotherapy group. Osimertinib was only discontinued in 7% of patients [37]. While there was a trend towards improved OS with osimertinib (26.2 vs. 22.5 months), this difference did not reach statistical significance (*p* = 0.277) [38].

### 2.2. Second- and Third-Generation EGFR TKIs versus First-Generation EGFR TKIs

The phase 2b LUX-Lung 7 clinical trial evaluated the use of afatinib vs. gefitinib in treatment-naïve patients with advanced *EGFR*-mutant NSCLC [19]. Afatinib conferred longer PFS (11 vs. 10.9 months; *p* = 0.017) and time-to-treatment failure (13.7 vs. 11.5 months; *p* = 0.0073) [19]. There were no differences in OS. The rate of toxicities and discontinuation of therapy were also similar (Table 1) [18,19]. In contrast, a real-world study showed that afatinib improved PFS and 1-year OS rates (16.4 months, *p =* 0.005; 78.2%; *p =* 0.004) compared to gefitinib (10.3 months and 69.1%) and erlotinib (12.1 months and 71.6%) [39].

Dacomitinib was approved in 2018 based on the phase 3 ARCHER 1050 trial [8,40,41]. In this study, first-line dacomitinib resulted in improved PFS (15 vs. 9 months; *p* < 0.0001) and OS (34 vs. 27 months; *p* = 0.438) compared to gefitinib [8,16]. The toxicities and rate of therapy discontinuation were similar in both groups (Table 1) [8,16].

Osimertinib was approved for the first-line treatment of *EGFR*-mutant NSCLC in 2018 based on results of the phase 3 FLAURA trial [7,36]. In this study, osimertinib demonstrated similar RR (80% vs. 76%) and disease control rate (DCR: 97% vs. 92%) to first-generation EGFR TKIS, but longer PFS (19 vs. 10 months; *p* < 0.001) and OS (39 vs. 32 months; *p* = 0.046) [7,17]. It was also better tolerated than gefitinib or erlotinib (Table 1) [7,17]. Based on these combined findings, osimertinib, afatinib and dacomitinib are preferred over first-generation EGFR TKIs in the front-line setting.

### 2.3. Second-Generation EGFR TKIs Following First-Generation EGFR TKI Failure

The phase 2 LUX-Lung 4 trial evaluated the use of afatinib in patients who progressed after treatment with erlotinib or gefitinib [42]. The RR was 8% and the PFS was 4.4 months. Approximately 37% of patients experienced a grade ≥3 toxicity and 29% of patients discontinued afatinib due to serious AEs [42]. Given the limited clinical efficacy, afatinib is not used in this setting.

### 2.4. Third-Generation EGFR TKIs Following First and Second-Generation EGFR TKI Failure

The phase 1/2 AURA trial evaluated the use of osimertinib after progression on gefitinib or erlotinib. The DCR and PFS were 84% and 8.2 months for all-comers, and 95% and 9.6 months for those with a T790M mutation, respectively [43]. The phase 2 AURA2 trial evaluated osimertinib in patients with *EGFR* T790M, who progressed on any first- or second-generation TKI. The DCR was 92%, RR was 70%, and PFS was 9.9 months. Approximately 34% of patients experienced a grade ≥3 toxicity but only 5% discontinued osimertinib due to an AE [44].

The GioTag observational study evaluated the role of sequencing osimertinib after afatinib failure in patients with *EGFR* T790M mutations [45]. The OS was 37.6 months for all-comers, 41.6 months for those with co-existing *EGFR* del19ex mutation, and 44.8 months for Asian patients [45]. A retrospective study from South Korea also demonstrated a role for sequencing osimertinib after afatinib among those with *EGFR* T790M-mutant NSCLC [46]. In this analysis, the median time on treatment for patients who received osimertinib was 20.8 months while the 2- and 3-year OS rates were 86% and 69%, respectively [46]. Therefore, osimertinib is recommended for patients with *EGFR* T790M-mutant NSCLC who progress after a first- or second-generation TKI.

### 2.5. EGFR TKIs in Combination with Anti-Vascular Endothelial Growth Factors

Combination therapies targeting the vascular endothelial growth factor (VEGF), or the VEGF receptor (VEGFR), and EGFR have been studied [47]. The phase 3 NEJ026 trial evaluated erlotinib +/− bevacizumab in Japanese patients with advanced *EGFR*-mutant NSCLC [48]. Erlotinib plus bevacizumab resulted in improved PFS (16.9 vs. 13.3 months; *p* = 0.016) without increasing toxicity rates [48]. Similarly, the phase 3 RELAY trial evaluating erlotinib +/− ramucirumab demonstrated an improved PFS in those receiving combination therapy (19 vs. 12 months; *p* < 0.0001) [49]. Grade ≥3 toxicities and treatment discontinuation, however, were more common in the combination therapy group (Table 1) [49]. While the median OS data is not yet available, there were no differences at 1 and 2 years between the two treatment groups [49].

Afatinib plus bevacizumab were evaluated in a phase 1 clinical trial in Japan achieving a RR of 81% [50]. Similarly, in an observational study from Taiwan using the same combination therapy the RR was 88%, the PFS was 24 months, and the OS was 46 months [51].

Osimertinib in combination with anti-VEGF has also been investigated. A phase 1/2 trial (NCT02803203) evaluated first-line osimertinib with bevacizumab [52]. The RR was 80% and the PFS was 19 months [52]. Approximately 31% of patients discontinued bevacizumab due to toxicity [52]. This combination was also evaluated in Japanese (WJOG 8715L) and European (BOOSTER) patients with *EGFR* T790M-mutant NSCLC who developed disease progression after a first- or second-generation EGFR TKI [53,54]. These phase 2 trials failed to demonstrate an improvement in PFS compared to osimertinib alone (14 vs. 9 months; *p* = 0.20 in the WJOG 8715L Trial, and 15 vs.12 months; *p* = 0.83 in the BOOSTER Trial) [53,54]. Combination therapy resulted in a significantly shorter time to treatment failure (8 vs. 11 months; *p* = 0.0074) and an increased incidence of grade ≥3 toxicities (47% vs. 18%) [54]. Another phase 2 trial (WJOG9717L) evaluated the use of osimertinib with bevacizumab in patients with common *EGFR* mutations [55]. The median PFS was similar to osimertinib alone (22.1 vs. 20.2 months; HR 0.862 *p* = 0.213), while the rate of grade ≥3 toxicities was higher in the combination group (56% vs. 48%) [55]. These combined findings suggest osimertinib plus bevacizumab does not improve outcomes and increases toxicities.

Currently, there are several ongoing studies evaluating bevacizumab or ramucirumab with second- or third-generation EGFR TKIs in the frontline setting (Clinicaltrials.gov: NCT04575415, NCT04148898, NCT03909334, NCT02971501, NCT04181060, accessed on 9 January 2023). The results of these studies are awaited and may affect current treatment paradigms.

### 2.6. EGFR TKIs in Combination with Chemotherapy

The use of EGFR TKIs with chemotherapy has been investigated in retrospective and prospective studies [56,57,58,59,60,61]. Although the results of these studies are mixed, some suggest that combination therapy may be more beneficial in selected treatment-naïve patients [56,57,58,59,60,61].

Erlotinib plus chemotherapy was evaluated as first-line therapy in patients with advanced/metastatic NSCLC with or without common *EGFR* mutation in the phase 2 trial CALGB 30406 [56]. This study included 66 patients with a common *EGFR* mutation. Of these, 33 received erlotinib monotherapy and 33 received erlotinib with paclitaxel and carboplatin. There were no significant differences in PFS for all-comers between the two treatment groups (PFS = 5.0 vs. 6.6 months; *p* = 0.1988) [56]. The phase 3 IMPRESS trial evaluated chemotherapy +/− gefitinib after disease progression on gefitinib [57]. The PFS was 5.4 months in both groups, however, more toxicities were seen in the combination arm [57]. Two phase 3 trials, one in India (CTRI/2016/08/007149) and one in Japan (NEJ009), evaluated gefitinib in combination with chemotherapy versus gefitinib alone in treatment-naïve patients [58,59]. The combination arms resulted in improved RR, longer PFS and OS. However, grade ≥3 toxicities were more frequent in the combination arms [58,59].

The LUX-Lung 5 trial evaluated afatinib plus chemotherapy vs. single agent chemotherapy in patients with advanced *EGFR*-mutant NSCLC who had progressed after treatment with first-generation EGFR TKI, chemotherapy, and afatinib [60]. In this phase 3 clinical trial, afatinib plus chemotherapy resulted in improved RR (32.1% vs. 13.2%; *p* = 0.005) and PFS (5.6 vs. 2.8 months; *p* = 0.003), however, there was no difference in OS between the two groups [60]. The incidence of grade ≥3 toxicities was higher in the combination group [60].

A retrospective study evaluated osimertinib plus chemotherapy in the third-line setting or beyond [61]. The median duration of treatment was 6.1 months, and the median OS was 10.4 months (95% CI 7.0–13.2 months). According to the authors, the OS was slightly inferior compared to the AURA3 trial likely reflecting that their populations was more heavily pre-treated than AURA3 [61]. Approximately 27% of patients developed grade ≥3 toxicities. The rate of osimertinib discontinuation was 2% [61]. Currently, the use of osimertinib in combination with chemotherapy is being investigated as first-line therapy in *EGFR*-mutant NSCLC in the phase 3 FLAURA2 trial (NCT04035486) and in patients with detectable *EGFR* mutations in ctDNA in two phase 2 studies (NCT04410796, NCT05281406). Preliminary data from FLAURA2 suggests that the combination is well tolerated and safe [62].

### 2.7. EGFR TKIs in Combination with Immunotherapy

The use of first-generation EGFR TKIs in combination with immune checkpoint inhibitors (ICIs) has been investigated in three phase 1 clinical trials in patients with *EGFR*-mutant NSCLC [63,64,65]. While response to therapy appeared promising, grade ≥3 toxicities were seen in more than 40% of patients and therapy was discontinued in 35% of patients [63,64,65].

Osimertinib was combined with durvalumab in one arm of the phase 1b TATTON trial [66]. The RR was 43% with this combination, however, the treatment arm was terminated because of increased reports of interstitial lung disease (ILD) [66]. The phase 3 CAURAL trial evaluated osimertinib with or without durvalumab in patients with *EGFR* T790M mutations who had received prior EGFR TKI [67]. The RR was 80% and no grade ≥3 toxicities were seen among the first 14 treated patients. However, recruitment was terminated due to the high rates of ILD reported in the TATTON trial [66,67]. While the combination of EGFR TKIs with ICIs appeared promising, the incidence and severity of AEs seems prohibitive.

## 3. Mechanisms of Resistance to EGFR TKIs

Cancer cells may develop innate or acquired resistance to EGFR TKI therapy (Table 2). Primary resistance occurs from coexisting uncommon *EGFR* mutations, mutations in genes other than *EGFR*, or heterogeneity to TKI response [68]. Acquired resistance is defined as progression on an EGFR TKI in a patient with a common *EGFR* mutation that achieved significant or durable (≥6 months) clinical response [69]. Acquired resistance mechanisms are further subclassified as *EGFR*-dependent and -independent, but these can coexist and overlap [70,71].

*EGFR*-dependent mechanisms lead to an increase in EGFR kinase activity. The most common is the development of the *EGFR* T790M mutation (“gatekeeper”) which accounts for 25–50% of cases of treatment failure after first- and second-generation EGFR TKIs [7,68,70,72,73,74,75,76,77,78,79]. The *EGFR* C797X mutations are also commonly seen. Specifically, the *EGFR* C797S mutation accounts for ~7–8% of cases of osimertinib resistance when used as first-line therapy and 10–26% of cases when used as second-line therapy [80,81]. Other mutations like *EGFR* L792X, G769X, L718Q, G719A, G724S, or exon 18 variants D761Y, S768I, V769L are less common [77,78,79,80,82,83].

*EGFR*-independent mechanisms include bypass mechanisms and histologic/phenotypic transformation [70,71,81,84,85,86,87,88,89]. EGFR proteins, as members of the ERBB/HER family, normally interact with other ERBB/HER family members to create dimers that phosphorylate and activate downstream signaling pathways [20]. Bypass mechanisms may be a result of *ERBB2/HER2* mutations/amplifications that form EGFR/HER2 dimers or active HER2 molecules with downstream activating effects [20]. *MET* amplifications can promote persistent HER3 tyrosine kinase activity with downstream activation [20]. Other bypass mechanisms of resistance may result from *PIK3CA*, *BRAF*, *KRAS*, and *MET* exon 14 skipping mutations, as well as *RET* and *FGFR3* fusions [84,85,86,87,88]. Upregulation of PD-L1 leading to immune escape has also been described [90]. Histologic transformation to small cell lung cancer (SCLC) occurs in 3–14% of NSCLC patients treated with EGFR TKIs [70,71,81,89]. It has been suggested that initial biopsies may fail to capture pre-existing SCLC and that treatment with EGFR TKIs results in regression of the NSCLC component while allowing the SCLC component to progress [91]. Patients with concurrent *EGFR/RB1/TP53* mutations seem to be at a particularly high risk of undergoing SCLC histologic transformation [92]. Epithelial-to-mesenchymal transition (EMT) is another mechanism of resistance affecting ~5% of EGFR-TKI resistant tumors [71]. EMT occurs following genetic changes in cancer cells that allow them to transition from having an epithelial to having a mesenchymal phenotype. This transition enables cancer cells to migrate, invade surrounding tissue, and become resistant to therapy [71,93,94].

## 4. Therapy Following EGFR TKI

The therapy choice after progression on EGFR TKIs varies according to symptoms, metastatic burden, mechanism of resistance, and the class of EGFR TKI used in the front-line setting. In patients who are initially treated with first- or second-generation EGFR TKIs and develop an *EGFR* T790M mutation, osimertinib is preferred over chemotherapy [37,38,44,61,95]. Patients who develop asymptomatic disease progression or oligoprogression (3 to 5 new metastasis) while on an EGFR TKI, but do not acquire an *EGFR* T790M mutation, should continue treatment with the same EGFR TKI plus local palliative therapy (surgery or radiation) to sites of active disease [96,97,98]. For symptomatic patients with multiple new metastases after an EGFR TKI without an *EGFR* T790M mutation, a change in systemic therapy is recommended (Table 3) [98,99,100,101,102,103]. The use of first- or second-generation TKIs after progression on osimertinib is not recommended given the poor disease control and short PFS [104].

Chemoimmunotherapy is considered the standard of care for patients with advanced NSCLC, but studies have often excluded patients with *EGFR* mutations [99,100,101]. The phase 3 trials IMpower130, IMpower150, and ORIENT-31, included patients with *EGFR*-mutant NSCLC after progression on EGFR TKIs [102,105,106,107]. IMpower150 evaluated the combination of carboplatin, paclitaxel, atezolizumab, and bevacizumab. This combination resulted in a trend towards improved RR (71% vs. 42%), PFS (10 vs. 7 months; HR 0.61–95% CI 0.36–1.03), and OS (26 vs. 20 months; HR 0.91–95% CI 0.53–1.59) compared to chemotherapy plus bevacizumab alone among patients with *EGFR*-mutant NSCLC; however, the differences were not statistically significant [105,106]. Further, there was no significant improvement in RR, PFS, or OS in the those receiving chemoimmunotherapy versus chemotherapy plus bevacizumab (Table 3) [106]. Similar results were seen in the IMpower130 trial, where the addition of immunotherapy to chemotherapy did not result in improved PFS or OS among patients with *EGFR*-mutant NSCLC [102]. The ORIENT-31 trial evaluated the combination of an anti-PD-1 agent (sintilimab), an anti-VEGF agent (IBI305), and chemotherapy in *EGFR*-mutant NSCLC after EGFR TKI failure [107]. Preliminary results demonstrated an improvement in PFS with chemoimmunotherapy plus anti-VEGF vs. chemotherapy alone (6.9 vs. 4.3 months; *p* < 0.0001). This improvement was also seen with the use of chemoimmunotherapy compared to chemotherapy alone (5.6 vs. 4.3 months; *p* < 0.0584) [107]. The safety and efficacy of chemoimmunotherapy in *EGFR*-mutant NSCLC is being assessed in ongoing phase 2 and 3 trials (NCT03786692, Checkmate 722–NCT02864251).

Expression of PD-L1 seems to play a role in response to immunotherapy alone in *EGFR*-mutant NSCLC [108,109,110]. A subgroup analysis in the KEYNOTE-001 trial suggested that patients with common *EGFR* mutations and PD-L1 ≥50% had better RR with pembrolizumab than those with PD-L1 <1% [108]. Similarly, the phase 2 ATLANTIC trial evaluating the use of durvalumab after ≥2 lines of therapy in those with *EGFR*-mutant NSCLC, demonstrated that patients with PD-L1 ≥25% had improved RR (12% vs. 4%) and OS (13.3 vs. 9.9 months) compared to those with PD-L1 <25% [109,110]. 

## 5. Novel EGFR TKIs and Targeted Therapies

Savolitinib, an oral TKI against c-MET, was combined with osimertinib in patients with *EGFR*-mutant NSCLC with *MET* amplifications/mutations after progression on osimertinib in the phase 2 trial ORCHARD [111]. The RR was 41% among the 17 evaluable patients, while PFS and OS was not reported [111]. The phase 1b TATTON trial evaluated the combination of osimertinib and savolitinib in patients who had progressed on osimertinib. Combination therapy resulted in a RR of 30% and a PFS of 5.4 months [112]. INSIGHT-2 is an ongoing phase 2 trial evaluating tepotinib (MET inhibitor) plus osimertinib in patients with *MET* amplification after progression on osimertinib (NCT03940703).

The phase 1 CHRYSALIS trial investigated amivantamab (bi-specific antibody against *EGFR* and *MET)* +/− lazertinib (third-generation EGFR TKI) in patients with *EGFR*-mutant NSCLC who progressed on osimertinib [113]. Preliminary results demonstrated a RR of 19% in the monotherapy group and 36% in those receiving combination therapy [113]. The phase 1 CHRYSALIS-2 trial evaluated amivantamab plus lazertinib in patients with *EGFR*-mutant NSCLC who progressed on osimertinib and platinum-based chemotherapy. Among the 50 evaluable patients, the RR was 36%, the median duration of response was not reached, and grade ≥3 toxicities mainly included infusion reactions, dermatitis, and hypoalbuminemia [114]. The amivantamab plus lazertinib combination has been moved to phase 3 investigation under the MARIPOSA trial. The latter is comparing frontline amivantamab plus lazertinib versus osimertinib alone in treatment-naïve patients with advanced NSCLC with common *EGFR* mutations (NCT04487080) [115].

Patritumumab deruxtecan, an anti-HER3 antibody-drug conjugate (ADC), was investigated in a phase 1 clinical trial (U3-1402) in patients with *EGFR*-mutant NSCLC who progressed on EGFR TKI [116]. The RR was 39% and the PFS was 8.2 months [116]. The phase 2 HERTHENA-Lung01 trial is underway to confirm these findings (NCT04619004).

Datopotamab deruxtecan, an ADC directed against Trop-2, was investigated in the phase 1 TROPION-PanTumor01 trial in patients with advanced NSCLC with an actionable mutation (including *EGFR*) who had previously progressed after treatment with a TKI and chemotherapy [117]. The RR was 35% and the median duration of response was 9.5 months [117]. The phase 3 TROPION-Lung01 trial is underway to confirm these findings (NCT04656652).

## 6. Special Considerations: Brain, Liver, and Bone Metastases

Retrospective data suggest that the metastatic pattern does not differ among patients with *EGFR*-mutant NSCLC compared to those without *EGFR* mutations [118]. Approximately 25% of patients with advanced *EGFR*-mutant NSCLC have brain metastases at the time of diagnosis and ~50% develop these within 3 years [119]. The management of brain metastasis may be challenging depending on the location and the number of lesions. Therefore, it is important to select an EGFR TKI with good CNS coverage. Unfortunately, many EGFR TKIs trials excluded patients with brain metastases [4,8,28,49]. A retrospective study assessing first-generation EGFR TKIs demonstrated that up to 12% of patients receiving first-line erlotinib and 18–30% of patients receiving first-line gefitinib developed CNS disease progression [120]. Findings from a retrospective Japanese study suggest that patients treated with erlotinib had a lower chance of developing CNS metastasis than those treated with gefitinib (4.8% vs. 24.5%; *p* = 0.04) [121].

The LUX-Lung 3 and LUX-Lung 6 trials included patients with asymptomatic brain metastases but did not report the rates of CNS progression [5,9]. This was reported in a cohort study in Taiwan [122]. Approximately 18% (N = 47) of patients treated with front-line afatinib developed CNS progression [122]. Among patients without known CNS metastasis, 11% developed brain metastases, while 33% of patients with known brain metastases had CNS disease progression [122]. The ARCHER 1050 study excluded patients with brain metastasis, however, 0.44% of patient treated with dacomitinib and 4.9% receiving gefitinib developed CNS disease [8,123]. In a series of 14 patients with brain metastases treated with first-line dacomitinib, nearly 86% had improvement of their CNS disease suggesting dacomitinib has CNS activity [124]. The CNS activity of dacomitinib is currently under investigation (NCT04675008).

The FLAURA trial allowed the enrollment of patients with neurologically stable CNS metastasis [7]. Approximately 6% of patients treated with osimertinib had progressive CNS disease versus 15% of patients treated with erlotinib or gefitinib [7]. In patients without known or treated CNS disease, 3% of patients on osimertinib and 7% receiving standard EGFR TKI developed CNS disease [125]. The CNS PFS was longer in those receiving osimertinib compared to standard EGFR TKI (median CNS PFS, not reached vs. 13.9 months; HR, 0.18; *p* = 0.014) [125]. The CNS RR in those with known brain metastases receiving osimertinib was better than in those receiving standard EGFR TKI (91% vs. 68%) [7,126].

Combination therapy may provide improved control of CNS disease. The RELAY study evaluating erlotinib +/− ramucirumab also excluded patients with brain metastases. In this study, however, only two patients (0.9%) treated with erlotinib plus ramucirumab developed CNS metastasis versus eight patients (3.6%) in the placebo plus erlotinib group [49]. It is unclear which approach between second-, third-generation EGFR TKIs, or combination anti-EGFR/VEGF therapy will result in better CNS outcomes. Osimertinib, however, remains the agent with the strongest evidence supporting its use in prevention and treatment of CNS metastasis.

Liver metastases affect 14–17% of patients with *EGFR*-mutant NSCLC [118,127]. This incidence is similar to that seen among *EGFR* wild-type NSCLC, suggesting *EGFR* mutations do not confer a higher risk for developing liver metastasis [128]. Although the treatment of patients with liver metastasis does not usually differ from patients without liver involvement, outcomes tend to be worse when liver metastases are present, even with the use of *EGFR* TKIs like osimertinib [128,129,130].

In contrast to liver, bone metastases occur more commonly in *EGFR*-mutant (40–54%) than in *EGFR* wild-type NSCLC (32%) [118,131]. Bone metastases seem to be associated with a lower risk of death among patients with *EGFR*-mutant NSCLC [132]. Furthermore, patients with *EGFR* mutations and bone metastases appear to have better OS than those without *EGFR* mutations and bone metastasis [132]. Retrospective data suggest that the use of osimertinib is associated with better clinical outcomes than the use of first- or second-generation *EGFR* TKIs in patients with this metastatic pattern [130]. Finally, the addition of bisphosphonates to therapy not only prevents skeletal complications but also seems to enhance the effect of EGFR TKIs and improve PFS [132].

## 7. Treatment Sequencing: A Suggested Approach

The ideal sequencing of EGFR TKIs and other therapies for patients with *EGFR*-mutant NSCLC remains uncertain. While osimertinib is often incorporated in the treatment of *EGFR*-mutant NSCLC, the timing of its incorporation remains unclear. The FLAURA trial demonstrated that frontline osimertinib conferred an advantage in PFS and OS when compared to first-generation EGFR TKIs [7]. After first- and second-generation TKI failure due to an acquired T790M mutation, osimertinib also improved outcomes compared to chemotherapy (AURA3 trial) [37]. However, it is unknown whether front-line osimertinib use results in more durable response than sequencing EGFR TKIs as in the AURA 1-3 trials [133,134,135].

Many experts advocate for the use of osimertinib in the front-line setting due to its better tolerability than first- and second-generation EGFR TKIs [134,135], and superior outcomes compared to first-generation EGFR TKIs [5,7,9,134]. However, other experts favor the use of osimertinib after progression on a first- or second-generation EGFR TKI to delay the use of chemotherapy [136,137]. The latter approach has some limitations. At the time of progression on first- or second-generation EGFR TKI, mutation analysis should be pursued, but testing for *EGFR* and other acquired mutations at progression may not be feasible or readily available [138,139]. According to the Flatiron Health database, evaluating a predominately US-based population, only 30% of patients were tested for *EGFR* mutations following progression on a first- or second-generation EGFR TKI [139]. Sequencing strategies also assume that patients who progress develop a T790M mutation. This mutation, however, only occurs in 25–50% of patients treated with a first- or second-generation EGFR TKIs [7,68,70,72,73,74,75,76,77,78,79]. A simulation study comparing first-line osimertinib to alternative EGFR TKI sequencing strategies suggests an improvement in PFS among those receiving osimertinib in the first-line setting regardless of the presence of a T790M mutation [135].

Sequencing EGFR TKIs also assumes patients will be fit to receive subsequent therapies, however, ~30% of patients will not be eligible for second-line treatment [139,140,141]. In the FLAURA trial, 35% of patients receiving first-generation EGFR TKIs did not receive second-line therapy [17]. A small retrospective real-world study conducted in three certified lung cancer centers in Germany also found that 30% of patients treated with front-line first- or second-generation EGFR TKIs did not receive second-line therapy due to poor performance status, CNS metastasis, rapid disease progression, or death [138]. Additionally, studies in the United States revealed that 28–30% of patients treated with frontline first- or second-generation EGFR TKI did not receive subsequent therapies [139,141]. Among those who received subsequent treatment, only 23% to 25% received osimertinib [139,141].

As it is difficult to predict the mechanisms of resistance and performance status at the time of disease progression, using upfront osimertinib over sequencing strategies may provide the patients the best outcomes for all-comers. If osimertinib is not available, a second-generation EGFR TKI is also a good choice, especially for those with uncommon *EGFR* mutations. The prospective APPLE study is evaluating the optimal strategy for osimertinib use (upfront vs. sequential) (NCT02856893) and hopefully will inform future EGFR TKIs sequencing.

The choice of upfront EGFR TKI may also be affected by the specific *EGFR* mutation. For example, in the LUX-Lung 3 and 6 trials, a subgroup analysis revealed superior OS in patients with EGFR ex19del receiving afatinib compared to chemotherapy [6]. However, there was no significant difference in OS in patients with L858R mutations receiving afatinib or chemotherapy [6]. In the FLAURA trial, patients receiving osimertinib who had an ex19del had improved OS compared to patients with L858R mutations [17]. In the ARCHER 1050 study, however, patients receiving dacomitinib who had L858R mutations only had a trend toward superior OS compared to patients with ex19del [16]. These combined findings suggest that ex19del mutations are associated with better prognosis.

The presence of CNS metastasis may also dictate the choice of front-line therapy. Osimertinib and afatinib are the only EGFR TKIs that have been evaluated in prospective trials in patients with CNS disease on presentation [5,7,9]. Osimertinib demonstrated superior CNS DCR and PFS compared to first-generation EGFR TKIs [7,121,125,126]. While there is no direct comparison with second-generation EGFR TKIs, preclinical data suggest that osimertinib provides improved CNS penetration compared to afatinib [142]. Approximately 11% of patients receiving front-line afatinib developed CNS metastases, while only 3% receiving front-line osimertinib developed CNS metastasis in the FLAURA trial [122,125].

## 8. Conclusions

Studies have demonstrated that the use of frontline second- and third-generation is preferred over first-generation EGFR TKIs in patients with *EGFR*-mutant NSCLC. Only osimertinib and dacomitinib have prospective data demonstrating improved PFS and OS over first-generation EGFR TKIs [16,17]. While there are no prospective studies to support an OS benefit of afatinib compared to first-generation EGFR TKIs, real-world evidence does suggest this benefit. Afatinib, osimertinib, and dacomitinib have not been compared head-to-head, therefore there is no strong evidence to support one over the other in the front-line setting. The use of erlotinib plus ramucirumab has also demonstrated good activity compared to erlotinib alone. However, OS data is lacking and patients with CNS metastasis were not included in this study. Furthermore, this approach is more costly and burdensome for patients as it involves infusions every 2 weeks.

When selecting a sequencing strategy for EGFR there are several aspects to consider, including CNS disease at presentation, the type of *EGFR* mutation at presentation, access to specific drugs, access to genomic testing at the time of progression, mechanisms of resistance, clinical performance status at progression, and provider level of comfort.

Evidence suggests osimertinib has better CNS penetration than afatinib and may provide better clinical outcomes in patients with bone metastasis, therefore we recommend it over other EGFR TKIs in patients with CNS or bone involvement. On the other hand, for patients with uncommon *EGFR* mutations, although osimertinib is an alternative, afatinib is the most extensively studied drug and the only one approved for this patient population [5,10,34,143].

When progression occurs and EGFR TKI therapy has been exhausted, a preferred therapeutic option is the use of chemoimmunotherapy with bevacizumab or the combination of chemotherapy plus bevacizumab, over chemotherapy alone. For patients who cannot receive an antiangiogenic agent, chemotherapy +/− EGFR TKI continuation should be considered. Immunotherapy alone could be another therapeutic option for selected patients with high PD-L1 levels (e.g., ≥25%) [109,110].

Finally, novel agents and drug combinations have shown promising results in early phase trials (Table 3). Ongoing studies evaluating the ideal sequencing and combination strategies to improve outcomes and overcome EGFR TKI resistance will hopefully inform the optimal treatment sequencing strategy (NCT04811001, NCT04413201, NCT04105153, NCT04035486, NCT03909334 and NCT02789345). Results of these studies are anxiously awaited.

## Figures and Tables

**Figure 1 cancers-15-00629-f001:**
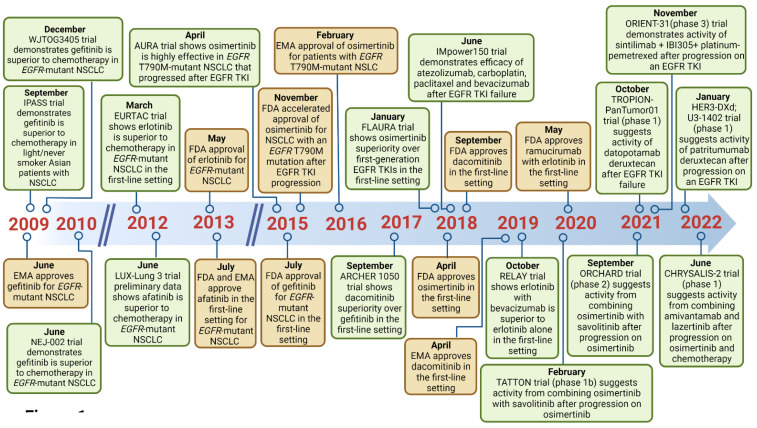
Timeline depicting the publication of landmark trials on anti-EGFR TKIs, approval by the FDA and EMA, and publication of data pertaining novel therapies; EMA, European Medicines Agency; FDA, Food and Drugs Administration; TKIs, tyrosine kinase inhibitors. Created with BioRender.com, accessed on 8 January 2023.

**Table 1 cancers-15-00629-t001:** Landmark Clinical Trials of EGFR TKIs as First-Line Therapy for Advanced NSCLC.

Drug(s)	Trial (NCT #)	Phase	Population Characteristics	Treatment Regimen /Cohorts	Outcomes	Side Effects	Rate of Discontinuation from Toxicity	FDA Approval Date
Erlotinib	EURTAC (NCT00446225)	III	N = 173 Stages IIIB/IV Adenocarcinoma only -Race: White (99%) -*EGFR* Mutations: ex19del and L858R -Asymptomatic brain metastases were allowed	Cohort A: Erlotinib 150 mg daily Cohort B: Platinum-based chemotherapy	ORR: 63.6% (A) vs. 17.8% (B) mPFS: 9.7 months (A) vs. 5.2 months (B) *p* < 0.0001 mOS: 19.3 months (A) vs. 19.5 months (B) *p* = 0.87	Overall grade ≥3 AE: 45% (A) vs. 67% (B)	13% (A) vs. 23% (B)	14 May 2013
Gefitinib	IPASS (NCT00322452)	III	N = 1217 Stage IIIB/IV Adenocarcinoma only -Race: Asian (98%) -*EGFR* Mutations: ex19del, L858R, T790M, other -Excluded patients with untreated brain metastases	Cohort A: Gefitinib 250 mg daily Cohort B: Platinum-based chemotherapy	ORR: 84.8% (A) vs. 43.2% (B) mPFS: not reported; HR, 0.48 (95% CI, 0.34 to 0.67) for those with *EGFR* mutation mOS: 18.8 months (A) vs. 17.4 months (B) *p* = 0.109	Overall grade ≥3 AE: not reported *	6.9% (A) vs. 13.6% (B)	13 July 2015
	NEJ-002 (N/A–Japan)	III	N = 230 Stage IIIB/IV NSCLC -Race: Asian/Japanese (100%) -*EGFR* Mutations: ex19del, L858R, and other (6.1%) -Asymptomatic brain metastases were allowed	Cohort A: Gefitinib 250 mg daily Cohort B: Platinum-based chemotherapy	ORR: 73.7% (A) vs. 30.7% (B) mPFS: 10.8 months (A) vs. 5.4 months (B) *p* < 0.001 mOS: 27.7 months (A) vs. 26.6 months (B) *p* = 0.483	Overall grade ≥3 AE: 41.2% (A) vs. 71.7% (B)	Not reported	
	WJTOG3405 (N/A–Japan)	III	N = 118 Stage IIIB/IV NSCLC -Race: Asian/Japanese (100%) -*EGFR* Mutations: ex19del and L858R -Asymptomatic brain metastases were allowed	Cohort A: Gefitinib 250 mg daily Cohort B: Platinum-based chemotherapy	ORR: 62.1% (A) vs. 32.2% (B) mPFS: 9.2 months (A) vs. 6.3 months (B) *p* < 0.0001 mOS: 34.9 months (A) vs. 37.3 months (B) *p* = 0.2070	Overall grade ≥3 AE: not reported *	Not Reported	
	IFUM (NCT01203917)	IV	N = 106 Stage IIIA/B/IV NSCLC -Race: White (100%) -*EGFR* Mutations: ex19del, L858R. T790M, S768I -Inclusion of brain metastatic disease not mentioned	Single-Arm: Gefitinib 250 mg daily	ORR: 69.8% mPFS: 9.7 months mOS: 19.2 months	Overall grade ≥3 AE: 15%	7.5%	
Afatinib	LUX-Lung 3 (NCT00949650)	III	N = 345 Stage IIIB/IV Adenocarcinoma only -Race: White (26.5%), Asian (71.7%), Other (1.7%) -*EGFR* Mutations: ex19del, L858R, Other (10.3%) -Asymptomatic stable brain metastases were allowed	Cohort A: Afatinib 40 mg daily Cohort B: Platinum-based chemotherapy	ORR: 56% (A) vs. 23% (B) mPFS: 11.1 months (A) vs. 6.9 months (B) *p* = 0.0004 mOS: 28.2 months (A) vs. 28.2 months (B) *p* = 0.39 mOS ex19del: 33.3 months (A) vs. 21.1 months (B) *p* = 0.0015 mOS L858R: 27.6 months (A) vs. 40.3 months (B) *p* = 0.29	Overall grade ≥3 AE: 49% (A) vs. 48% (B)	8% (A) vs. 12% (B)	Approval for *EGFR* exon 19 deletions or exon 21 (L858R): 23 July 2013 Expansion of indication to all non-resistant *EGFR* mutations: 12 January 2018
	LUX-Lung 6 (NCT01121393)	III	N = 364 Stage IIIB/IV Adenocarcinoma only -Race: Asian 100% -*EGFR* Mutations: ex19del, L858R, other (11%) -Asymptomatic, stable brain metastases were allowed	Cohort A: Afatinib 40 mg daily Cohort B: Platinum-based chemotherapy	ORR: 66.9% (A) vs. 23% (B) mPFS: 13.7 months (A) vs. 5.6 months (B) *p* < 0.0001 mOS: 23.1 months (A) vs. 23.5 months (B) *p* = 0.61 mOS ex19del: 31.4 months (A) vs. 18.4 months (B) *p* = 0.023 mOS L858R: 19.6 months (A) vs. 24.3 months (B) *p* = 0.34	Overall grade ≥3 AE: 36% (A) vs. 60.2% (B)	5.9% (A) vs. 39.8% (B)	
	LUX-Lung 7	IIb	N = 319 Stage IIIB/IV Adenocarcinoma only -Race: Asian (59%), White (30%), Black (1%), not available (11%) -*EGFR* Mutations: ex19del and L858R -Active brain metastases (symptomatic or requiring treatment) excluded	Cohort A: Afatinib 40 mg daily with escalation to 50 mg daily if well tolerated after 4 weeks Cohort B: Gefitinib 250 mg daily	ORR: 70% (A) vs. 56% (B) mPFS: 11.0 months (A) vs. 10.9 months (B), HR 0.73 *p* < 0.017 mOS: 27.9 months (A) vs. 24.9 months (B) *p* = 0.258	Overall grade ≥3 AE: 57% (A) vs. 52% (B)	6% (A) vs. 6% (B)	
Dacomitinib	ARCHER 1050 (NCT01774721)	III	N = 452 Stage IIIB/IV NSCLC -Race: Asian (75%), Black (<1%), White (25%) -*EGFR* Mutations: ex19del, L858R -Brain or leptomeningeal metastases excluded	Cohort A: Dacomitinib 45 mg daily Cohort B: Gefitinib 250 mg daily	ORR: 75% (A) vs. 72% (B) mPFS: 14.7 months (A) vs. 9.2 months (B) *p* < 0.0001 mOS: 34.1 months (A) vs. 26.8 months (B) *p* = 0.438	Overall grade ≥3 AE: 63% (A) vs. 41% (B)	10% (A) vs. 7% (B)	27 September 2018
Osimertinib	FLAURA (NCT02296125)	III	N = 556 Stage IIIB/IV NSCLC -Race: Asian (62%), White (36%), Other (1%) -*EGFR* Mutations: ex19del, L858R -Asymptomatic, stable brain metastases were allowed	Cohort A: Osimertinib 80 mg daily Cohort B: Gefitinib 250 mg daily or Erlotinib 150 mg daily	ORR: 80% (A) vs. 76% (B) mPFS: 18.9 months (A) vs. 10.2 months (B) *p* < 0.001 mOS: 38.6 months (A) vs. 31.8 months (B) *p* = 0.046	Overall grade ≥3 AE: 42% (A) vs. 47% (B)	15% (A) vs. 18% (B)	18 April 2018
Erlotinib + ramucirumab	RELAY (NCT02411448)	III	N = 449 Stage IV NSCLC -Race: Asian (77%), White (22.3%), Other (1%) -*EGFR* Mutations: ex19del, L858R -Brain or leptomeningeal metastases excluded	Cohort A: Erlotinib 150 mg daily + ramucirumab 10 mg/kg once every 2 weeks Cohort B: Erlotinib 150 mg daily + placebo once every 2 weeks	ORR: 76% (A) vs. 75% (B) mPFS: 19.4 months (A) vs. 12.4 month *p ≤* 0.0001 mOS: Not available	Overall grade ≥3 AE: 72% (A) vs. 54% (B)	13% (A) vs. 11% (B)	29 May 2020

* Only individual grade ≥3 AEs are reported; Abbreviations: AEs, adverse events; FDA, Food and Drug Administration; mPFS, median progression-free survival; mOS, median overall survival; ORR, overall response rate; NSCLC, non-small cell lung cancer; TKI, tyrosine kinase inhibitor.

**Table 2 cancers-15-00629-t002:** Mechanisms of Resistance to EGFR-TKI.

Classification	Sub-Classification	Examples
Primary	Coexisting Activating Mutations/fusions	Uncommon *EGFR* Mutations: *EGFR* Exon 20 insertions or duplications, de novo *EGFR* T790M Other: *MET* amplifications, *ALK* fusions/*EML4-ALK* fusions
	Heterogeneity in TKI Response	Cellular apoptotic machinery heterogeneity/Baseline BIM protein expression differences
Secondary/ Acquired	*EGFR*-Dependent	*EGFR* T790M (“gatekeeper” mutation) Non-T790M *EGFR* Mutations: D761Y, S768I, V769L, C797X, L792X, G719A, G769X, L718Q, or G724S
	*EGFR*-Independent	Bypass Mechanisms: (A) Genetic alterations: *MET* exon 14 skipping mutation, *ERBB2/HER2* mutations/amplification, *HER3* upregulation, *RET* or *FGFR3* fusions, *PIK3CA/BRAF*/*KRAS* mutations (B) Immune escape: PD-L1 upregulation Histologic Transformation: (A) Small Cell Lung Cancer (B) EMT

Abbreviations: EMT: Epithelial-to-mesenchymal transformation; PD-L1: Programmed Death-Ligand 1; TKI, tyrosine kinase inhibitor.

**Table 3 cancers-15-00629-t003:** Strategies to Overcome TKI Resistance.

Strategy	Drugs	Supporting Clinical Trial	Population	Intervention	ORR	PFS	OS
3rd Generation TKIs	Osimertinib	AURA 3	Stages IIIB/IV Adenocarcinoma *EGFR* T790M after failure to 1st or 2nd generation TKIs N = 419	Cohort A: Osimertinib 80 mg daily Cohort B: Pemetrexed 500 mg/m^2^ with either carboplatin AUC 5 or cisplatin 75 mg/m^2^ every 3 weeks	71% (A) vs. 31% (B) *p* < 0.001	10.1 months (A) vs. 4.4 months (B), HR 0.30 *p* < 0.001	26.8 months (A) vs. 22.5 months (B) *p* = 0.277
Chemo-Immunotherapy +/− anti-VEGF therapy	Atezolizumab +/− Bevacizumab	IMpower150	Stage IV non-squamous NSCLC. Those with *EGFR* mutations should have received and progressed or had unacceptable toxicities while on TKI N = 124/1202 *EGFR* positive	Cohort A: ABCP: Atezolizumab 1200 mg, bevacizumab 15 mg/Kg, carboplatin AUC 6, paclitaxel 200 mg/m^2^ every 3 weeks Cohort B: ACP: Atezolizumab 1200 mg, carboplatin AUC 6, paclitaxel 200 mg/m^2^ every 3 weeks Cohort C: BCP: Bevacizumab 15 mg/Kg, carboplatin AUC 6, paclitaxel 200 mg/m^2^ every 3 weeks	70.6% (A) vs. 35.6% (B) vs. 41.9% (C)	10.2 months (A) vs. 6.9 months (C) HR 0.61 CI 0.36–1.03 6.9 months (B) vs. 6.9 months (C), HR 1.14 CI 0.73–1.78	26.1 months (A) vs. 20.3 months (C), HR 0.91 CI 0.53–1.59 21.4 months (B) vs. 20.3 months (C), HR 1.16 CI 0.71–1.89
		IMpower130	Stage IV non-squamous NSCLC. Those with *EGFR* mutations should have received and progressed or had unacceptable toxicities while on TKI N = 44/724 with *EGFR* or *ALK* genomic aberrations	Cohort A: Atezolizumab 1200 mg, carboplatin AUC 6, and nab-paclitaxel 100 mg/m^2^ every 3 weeks Cohort B: Carboplatin AUC 6, and nab-paclitaxel 100 mg/m^2^ every 3 weeks	*EGFR*-cohort not reported	7.0 months (A) vs. 6.0 months (B), HR 0.75 CI 0.36–1.54	14.4 months (A) vs. 10.0 months (B), HR 0.98 CI 0.41–2.31
	Sintilimab +/− IBI305	ORIENT-31	*EGFR*-mutant non-squamous NSCLC who had progressed after EGFR TKI N = 444	Cohort A: Sintilimab 200 mg, IBI305 15 mg/Kg, cisplatin 75 mg/m^2^, pemetrexed 500 mg/m^2^ every 3 weeks Cohort B: Sintilimab 200 mg, placebo, cisplatin 75 mg/m^2^, pemetrexed 500 mg/m^2^ every 3 weeks Cohort C: Placebo, cisplatin 75 mg/m^2^, pemetrexed 500 mg/m^2^ every 3 weeks	43.9% (A) vs. 33.1% (B) vs. 25.2% (C)	6.9 months (A) vs. 4.3 months (C) HR 0.464 *p* < 0.0001 5.6 months (B) vs. 4.3 months (C) HR 0.726 *p* = 0.0584	NA
Immunotherapy	Pembrolizumab	KEYNOTE-001	Advanced NSCLC N = 78/550 with common *EGFR* mutations	Pembrolizumab 2 or 10 mg/Kg every 3 weeks, or 10 mg/Kg every 2 weeks	7.7% (all), 20% PD-L1 ≥50%, 8.7% PD-L1 1–49%, 0% <1%	NA	NA
	Durvalumab	ATLANTIC	Advanced NSCLC and disease progression after ≥2 systemic therapies N = 111/444 with *EGFR* or *ALK* mutations	Durvalumab 10 mg/Kg every 2 weeks	3.6% (PD-L1 <25%) vs. 12.2% PD-L1 ≥25%	1.9 months (PD-L1 <25%) vs. 1.9 months (PD-L1 ≥25%)	9.9 months (PD-L1 <25%) vs. 13.3 months (PD-L1 ≥25%)
c-MET Agents		ORCHARD (Experimental Module 1)	Metastatic NSCLC with *EGFR* and *MET* alterations after progression on first line osimertinib N = 20	Osimertinib 80 mg daily with savolitinib 300 or 600 mg daily	41%	NA	NA
		TATTON	Locally advanced or metastatic NSCLC with *EGFR* mutation and *MET* amplification after progression on EGFR TKIs Cohort B1: Previously treated with 3rd generation TKI N = 69 Cohort B2: No 3rd previous generation TKI T790M negative N = 51 Cohort B3: No 3rd previous generation TKI T790M positive N = 18 Cohort D: No previous 3rd generation TKI T790M negative N = 42	Cohort B: Osimertinib 80 mg daily with savolitinib 300 mg (if ≤55 Kg) or 600 mg daily Cohort D: Osimertinib 80 mg daily with savolitinib 300 mg daily	All B: 48% B1: 30% B2: 65% B3: 67% D: 64%	All B: 7.6 months B1: 5.4 months B2: 9.0 months B3: 11.0 months D: 9.1 months	NA
		CHRYSALIS	Metastatic or unresectable NSCLC with *EGFR* mutation who progressed on osimertinib and were chemotherapy naïve N = 166 (121 cohort A, 45 cohort B)	Cohort A: Amivantamab 1050 mg (1400 mg for patients ≥80 kg) weekly Cohort B: Amivantamab 1050 mg (1400 mg for patients ≥80 kg) weekly with lazertinib 240 mg daily	Cohort A: 19% Cohort B: 36%	NA Cohort A median DOR: 5.9 months Cohort B median DOR: 9.6 months	NA
		CHRYSALIS 2	Advanced or metastatic NSCLC with *EGFR* exon 19 deletion or L858R that progressed after osimertinib (1st or 2nd line) and platinum-based chemotherapy N = 162	Amivantamab 1050 mg (1400 mg if ≥80 Kg) with lazertinib 240 mg daily	36%	NA DOR: not reached	NA
ADC		NCT03260491	Locally advanced or metastatic *EGFR*-mutant NSCLC who fail prior TKI N = 57 Prior osimertinib and platinum-based chemotherapy: N = 44	HER3-DXd (pertuzumab deruxtecan) 5.6 mg/kg every 3 weeks	All: 39% Prior osimertinib and platinum chemotherapy: 39%	All: 8.2 months Prior osimertinib and platinum chemotherapy: 8.2 months	NA
		TROPION-PanTumor 01	Advanced or metastatic NSCLC with Actionable Mutations who failed TKI and chemotherapy N = 34 *EGFR*-mutant: N = 29, 65% after osimertinib	Datopotamab deruxtecan 4 mg/Kg, 6 mg/Kg, or 8 mg/Kg	35%	NA Median DOR: 9.5 months	NA

Abbreviations: ADC, Antibody Drug Conjugate; AEs, adverse events; AUC, area under the curve; DOR, duration of response; NA, not available; NSCLC, non-small cell lung cancer; OS, overall survival; ORR, overall response rate; PFS, progression-free survival; TKI, tyrosine kinase inhibitor.

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
