# Peer review of "Treatment Strategies for Non-Small Cell Lung Cancer with Common EGFR Mutations: A Review of the History of EGFR TKIs Approval and Emerging Data"

_cancers, 2023, doi:10.3390/cancers15030629_

Round 1
Reviewer 1 Report
Major points
1) At the line 84, “Erlotinib was the first EGFR TKI approved for unselected patients with advanced NSCLC in 2004 [22, 23]” is cited. However, gefitinib was the first EGFR TKI approved in the world, in Japan, in 2002. If the authors intend to describe about US approval, this should be cited.
2) From the line 92 to the 98, some explanations should be added. The Iressa Pan-Asia Study (IPASS), which compared gefitinib and carboplatin/paclitaxel in previously untreated never-smokers and light ex-smokers with advanced pulmonary adenocarcinoma, revealed that the objective response rate in the overall population was significantly higher with gefitinib than with carboplatin–paclitaxel (43.0% vs. 32.2%; odds ratio, 1.59; 95% CI, 1.25 to 2.01; P<0.001). In the subgroup analysis, the objective response rate was 71.2% with gefitinib versus 47.3% with carboplatin–paclitaxel in the mutation-positive subgroup (P<0.001) and 1.1% (one patient) versus 23.5%, respectively, in the mutation-negative subgroup (P=0.001).
In post hoc analyses, PFS was significantly longer for gefitinib versus carboplatin/paclitaxel in both the exon 19 deletions (HR, 0.38; 95% CI, 0.26 to 0.56) and the exon 21 L858R mutation (HR, 0.55; 95% CI, 0.35 to 0.87) subgroups. ORR was significantly higher with gefitinib (84.8%) versus carboplatin/paclitaxel (43.2%; OR, 7.23; 95% CI, 3.19 to 16.37) in the exon 19 deletions subgroup and higher (but not statistically significant) in the L858R subgroup (60.9% v 53.2%; OR, 1.41; 95% CI, 0.65 to 3.05). The second sentence in this paragraph gave readers information about only exon 19 deletion.
Any scientistic clinical research needs a statistical assumption in the primary endpoint. In this sense, IPASS lacked a statistical assumption in the primary endpoint in terms of EGFR mutation positive or negative. First clinical studies having the primary endpoint in terms of EGFR mutation positive or negative were NEJ002, WJTOG3405, EURTAC-SLCG GECP06/01, and OPTIMAL (CTONG 0802) studies. Referring to NEJ002 and WJTOG3405, National Institute for Health and Clinical Excellence (NICE) approved gefitinib for the first-line treatment of locally advanced or metastatic NSCLC with EGFR mutations (#192) in 2010. In #192, it was emphasized that “Testing for EGFR mutations should be carried out on all eligible patients irrespective of their gender, ethnicity, and smoking status to ensure that all patients who could benefit from gefitinib are identified.”
3) Therefore, Figure 1 should be modified.
4) In a review article, objective explanations and authors opinions should be divided. The latter should be cited in the last part of the review.
(a) at line 332, we recommend --- should be moved to the last part.
(b) the content from line 408 to 471 should be moved to the last part.
Reviewer 2 Report
The manuscript entitled “Treatment Strategies for Non-Small Cell Lung Cancer with Common EGFR Mutations: A Review of the History of EGFR TKIs Approval and Emerging Data” is suitable for publication in ‘Cancers’ after major revision.
Comments
- In the sentence “In the past two decades, five EGFR tyrosine kinase inhibitors (TKIs) have become commercially available for the management of advanced NSCLC with common EGFR-sensitizing mutations [i.e., EGFR exon 19 deletions or exon 21 mutations (L858R)] (Table 1; Figure 1) [3-9].” Authors should write the name of EGFR tyrosine kinase inhibitors instead of “five”.
- Authors should give detail about “platinum-doublet chemotherapy” in Line 68.
- Authors should discuss also general metastasis not only brain metastases in “Special Considerations: Brain Metastases” in Line 369.
- One of the resistance of EGFR in lung cancer, it is dimerization of EGFR with HER2. Authors should discuss this the connection of EGFR with other ErbB family members of receptor tyrosine kinases.
Round 2
Reviewer 1 Report
In fourth paragraph (line 560-567) of "in conclusion," the use of chemoimmunotherapy with antiangiogenic therapy is controversies for these patients. Therefore, instead of “recommended”, the expression of “a preferred therapeutic option” should be used. Do you have any evidence of “Immunotherapy alone can also be considered in frail patients with high PD-L1 levels (e.g., ≥25-50%) who cannot tolerate chemotherapy.”? The term of “a preferred therapeutic option” might be also used in this point.
As a minor point, “chemotherapy plus bevacizumab” is double typed?
Reviewer 2 Report
Authors have done significant changes.
Author Response
Authors have done significant changes.
We are grateful to the reviewer for their comment and the detailed analysis of our manuscript.
